# Foliar Application of Sodium Nitroprusside Boosts *Solanum lycopersicum* L. Tolerance to Glyphosate by Preventing Redox Disorders and Stimulating Herbicide Detoxification Pathways

**DOI:** 10.3390/plants10091862

**Published:** 2021-09-09

**Authors:** Cristiano Soares, Francisca Rodrigues, Bruno Sousa, Edgar Pinto, Isabel M. P. L. V. O. Ferreira, Ruth Pereira, Fernanda Fidalgo

**Affiliations:** 1GreenUPorto—Sustainable Agrifood Production Research Centre & INOV4AGRO, Biology Department, Faculty of Sciences of University of Porto, Rua do Campo Alegre s/n, 4169-007 Porto, Portugal; franciscamrodrigues98@gmail.com (F.R.); bruno.filipe@fc.up.pt (B.S.); ruth.pereira@fc.up.pt (R.P.); ffidalgo@fc.up.pt (F.F.); 2LAQV/REQUIMTE, Laboratory of Bromatology and Hydrology, Department of Chemical Sciences, Faculty of Pharmacy, University of Porto (FFUP), Rua de Jorge Viterbo Ferreira nº 228, 4050-313 Porto, Portugal; ecp@ess.ipp.pt (E.P.); isabel.ferreira@ff.up.pt (I.M.P.L.V.O.F.); 3Department of Environmental Health, School of Health, P.Porto (ESS-P.Porto), Rua Dr. António Bernardino de Almeida, 400, 4200-072 Porto, Portugal

**Keywords:** herbicides, non-target toxicity, redox homeostasis, stress alleviation, antioxidant system

## Abstract

Strategies to minimize the effects of glyphosate (GLY), the most used herbicide worldwide, on non-target plants need to be developed. In this context, the current study was designed to evaluate the potential of nitric oxide (NO), provided as 200 µM sodium nitroprusside (SNP), to ameliorate GLY (10 mg kg^−1^ soil) phytotoxicity in tomato plants. Upon herbicide exposure, plant development was majorly inhibited in shoots and roots, followed by a decrease in flowering and fruit set; however, the co-application of NO partially prevented these symptoms, improving plant growth. Concerning redox homeostasis, lipid peroxidation (LP) and reactive oxygen species (ROS) levels rose in response to GLY in shoots of tomato plants, but not in roots. Additionally, GLY induced the overaccumulation of proline and glutathione, and altered ascorbate redox state, but resulted in the inhibition of the antioxidant enzymes. Upon co-treatment with NO, the non-enzymatic antioxidants were not particularly changed, but an upregulation of all antioxidant enzymes was found, which helped to keep ROS and LP under control. Overall, data point towards the benefits of NO against GLY in tomato plants by reducing the oxidative damage and stimulating detoxification pathways, while also preventing GLY-induced impairment of flowering and fruit fresh mass.

## 1. Introduction

Glyphosate (GLY; N-(phosphonomethyl)glycine), the active compound of several commercial herbicides, was introduced on the pesticide market by Monsanto Company (S.A., Belgium, Europe) in the mid-1970s and has been in a leading position since then [1,2,3]. As a broad-spectrum herbicide, GLY’s use was initially restricted for weed removal from cultivated fields, meadows and non-crop areas [2]. However, since 1996, the introduction of transgenic GLY-resistant crops has led to a general upward trend of GLY-based herbicides application [1]. Indeed, currently, GLY is the most applied herbicide worldwide, accounting, in 2014, for more than 90% of the total herbicide market targeting the agricultural sector [4].

Paired with this increasing popularity, emerging concerns on GLY accumulation across the environment have begun to arise. With effect, it has been reported that this agrochemical can accumulate in soil due to leaching losses through the action of rain and/or wind during and after foliar application [3,5,6]. Moreover, once applied to weeds foliage, GLY can be translocated to the roots and gradually released, leading to its accumulation in the rhizosphere [6,7]. When in soil, residual amounts of GLY can then affect non-target plant species [8,9] since, even upon its metabolization by microorganisms and/or adsorption to soil components, the byproduct of its degradation, aminomethylphosphonic acid (AMPA), is also a recognized phytotoxin [3,8]. 

Once taken up by plants, GLY is promptly transported to meristems, young roots and leaves, storage organs and any other actively growing tissues through xylem and phloem loading [1]. In terms of action, GLY acts by inhibiting the activity 5-enolpyruvylshikimate-3-phosphate synthase (EPSPS; EC 2.5.1.19), consequently blocking the shikimate pathway involved in the biosynthesis of phenolic compounds and essential aromatic amino acids, such as phenylalanine, tyrosine and tryptophan [3]. Moreover, aside from its primary target effect, increasing evidence has suggested that this herbicide can induce oxidative bursts in plant cells, while also affecting the uptake of essential nutrients [3]. Thus, as there is a high demand for agriculture to exponentially increase food production, it is imperative to develop sustainable approaches to increase crops’ tolerance to GLY contamination.

Nitric oxide (NO), due to its small size and ability to easily diffuse across biological membranes, is recognized as a remarkable signalling molecule involved in the response of plants to different environmental constraints [10]. In fact, numerous studies conducted with several plant models have been pointing towards the important role of NO as an ameliorative agent against abiotic stresses [10,11,12,13,14,15]. Accordingly, the exogenous application of NO may result in an enhanced crop yield under adverse conditions, due to its role in regulating mechanisms related to increased tolerance to abiotic stress [14]. One of the most commons ways to study NO-mediated effects on plants is through the exogenous application of chemical donors, such as sodium nitroprusside (SNP). Chemically, it is an inorganic molecule composed of Fe (II) and NO^+^, being a derivate from iron-nitrosyl compounds [16,17]. When in solution, SNP releases NO^+^, Fe (II) and cyanide (CN−), which can sometimes mask the effects of NO [18]. Either way, this molecule, compared to others, has a relatively lower cost and is recognized for allowing a continuous and enduring production of NO [17,19]. Even though NO is a gaseous reactive nitrogen species (RNS), it has the ability to limit reactive oxygen species (ROS)-induced damages by acting as a chain breaker and by activating gene expression of antioxidant enzymes [10,11,12]. The involvement of NO in enhancing the antioxidant network in plants is well described in the literature and strongly suggests that NO-mediated increase of plant abiotic stress tolerance is related to a greater ROS detoxification by defence mechanisms [15]. Additionally, NO itself is known to have antioxidant properties, being involved in ROS detoxification and subsequently helping in the inhibition of lipid peroxidation (LP) and protein oxidation [20]. Despite the role of NO is relatively well understood in situations of drought, salinity and metal contamination [15,21,22,23], its involvement in herbicide-induced phytotoxicity, including GLY, remains poorly explored. Regarding this matter, only a recent study conducted by Singh et al. [24] is available, in which the potential of this RNS to alleviate GLY-induced stress in *Pisum sativum* L. was evaluated. In spite of the positive outcomes, this study only focused on the early development of seedlings (7 days old) and applied a high concentration of GLY (40 mg L^−1^) under a hydroponic system, not mimicking a real scenario of soil contamination. Moreover, the precise involvement of NO on the interplay between plant growth and productivity, GLY bioaccumulation and the modulation of antioxidant and detoxification pathways is yet to be uncovered. 

Within this perspective, and as previous studies from our research group have shown that soil contamination by GLY can negatively affect the growth and physiology of non-target plant species, such as tomato (*Solanum lycopersicum* L.) [8,25] and barley (*Hordeum vulgare* L.) [9], the main objectives of this study were (i) to evaluate the potential protective role of NO in counteracting GLY-induced stress in crops; and (ii) to pinpoint the main physiological and biochemical mechanisms behind NO action in GLY-exposed plants. Since *S. lycopersicum* (tomato) is one of the most important species worldwide and has been widely used as a model organism for fleshy-fruited plants [26], this species was selected for this study.

## 2. Results

### 2.1. Biometric Analysis—Fresh Biomass and Root Length

The presence of soil residues of GLY inhibited plant growth, as evidenced by a significant decrease in root length (49%), and fresh biomass of roots and shoots (73% and 48%, respectively), in relation to the CTL (Figure 1). However, after co-exposure to NO, GLY phytotoxic effects were partially prevented in all growth-related parameters, especially when root fresh biomass (107% increase when compared to the GLY treatment) is concerned. This NO-mediated increase in root growth was also noticed when plants were treated only with this molecule, with significant rises up to 65% in relation to the CTL. 

### 2.2. Soluble Protein and Nitrate Reductase (NR) Activity

Results concerning total protein content and NR activity are shown in Table 1 and Table 2. As can be observed, in the shoots, GLY led to a significant increase in protein levels (27%), regardless of the co-exposure to NO. Nevertheless, in the roots, herbicide treatment resulted in decreased protein levels by 50%, in relation to the CTL, being this effect significantly counteracted by the foliar application of NO (Table 2). Concerning NR, its activity significantly decreased in shoots among treatments, with inhibition values up to 40% compared to the CTL; in the roots, only plants co-exposed to GLY and NO showed a decline in the activity of this enzyme by 24% and 37%, in relation to the CTL and to the plants exposed to GLY alone, respectively (Table 1 and Table 2).

### 2.3. Biomarkers of Oxidative Stress

#### 2.3.1. Superoxide Anion (O_2_^•−^) and Hydrogen Peroxide (H_2_O_2_)

O_2_^•−^ levels were enhanced in shoots (75%) and roots (81%) of plants exposed to GLY (Figure 2a,d), compared to CTL. With the simultaneous application of NO, the levels of this ROS showed a significant decrease of 74% in shoots and 55% in roots, in relation to the GLY treatment; in shoots, O_2_^•−^ content from GLY + NO plants were even lower than those found in the CTL (decrease of 55%). Regarding H_2_O_2_, differences were detected only in the roots, where plants grown in GLY-contaminated soil, but treated with NO, experienced a sharp reduction over the CTL (44%) and GLY (36%) groups (Figure 2b,e). 

#### 2.3.2. Malondialdehyde (MDA) Content

LP, evaluated in terms of MDA content, was diminished by 37% in roots and increased by 33% in shoots upon GLY single exposure. In response to NO co-application, MDA levels were restored back to the levels found in the CTL (Figure 2c,f). 

### 2.4. Evaluation of the Non-Enzymatic Antioxidant Response

#### 2.4.1. Ascorbate (AsA), Glutathione (GSH) and Proline

Total AsA levels in shoots exhibited a tendency to increase in response to GLY, especially under NO co-exposure, where a significant rise of 49% compared to the CTL was recorded (Table 1 and Table 2). In roots, total AsA levels did not vary among treatments, with the exception of NO-treated plants, which showed an increment of 59% in relation to the CTL (Table 1 and Table 2). Concerning the ratio between AsA and DHA, in shoots, only NO promoted a significant decrease (87%) of this parameter, though GLY-treated plants also showed a tendency to have reduced values of AsA/DHA by 50%; in the roots, a significant increase of 44% of this ratio was found, over the CTL, when plants were exposed to GLY but simultaneously treated with NO (Table 1 and Table 2). 

The results of GSH accumulation are presented in Table 1 and Table 2. As shown, GLY-treated plants present increased levels of this antioxidant in shoots (58%) and roots (102%), in relation to the CTL. The co-application of NO did not significantly alter this response in the shoots; however, in the roots, the GSH content was restored to that found in the CTL. 

Concerning proline levels, plants’ response to GLY was similar in shoots and roots (Table 1 and Table 2). As can be observed, proline was severely increased in both organs (1-fold in roots and 4.3-fold in shoots), but the co-treatment with NO was able to inhibit this effect, since no significant differences were registered in relation to the CTL (Table 1 and Table 2).

#### 2.4.2. Total Phenolic Content (TPC), Flavonoids and Total Antioxidant Capacity (TAC)

In shoots, all treatments led to significantly lower levels of total phenolics, in comparison to the CTL (Table 1); in roots, however, their content did not vary among treatments (Table 2). Flavonoids, as shown in Table 1 and Table 2, followed the same trend of TPC, being overall diminished in response to GLY and/or NO, in shoots, and showing no variations in roots. The TAC values, compiled in Table 1 and Table 2, presented a similar pattern to that found for TPC with a general decrease in the roots of tomato plants exposed to GLY (inhibition around 33%), regardless of the co-application of NO, and with no major changes in the shoots. Even so, when GLY-exposed plants were sprayed with NO, TAC was only 16% lower than the CTL (Table 1 and Table 2).

### 2.5. Antioxidant Enzymes Activity (Superoxide Dismutase (SOD, EC 1.15.11); Glutathione S-Transferase (GST, EC 2.5.1.18); Catalase (CAT, EC 2.5.1.18); Ascorbate Peroxidase (APX, EC 1.11.1.11))

Data reporting SOD, GST, APX and CAT total activities are presented in Figure 3 and Figure 4. As shown, SOD was only significantly altered in the roots by exposure to GLY alone, where a 41% inhibition was found in relation to the CTL plants (Figure 3a,c). GST activity was also substantially reduced in both shoots (30%) and roots (58%) upon exposure to the herbicide. In response to the co-application of NO, GLY-exposed plants exhibited higher activity values of this enzyme in the roots and, especially, in the shoots, without differences from the CTL situation (Figure 3b,d). 

APX activity suffered a significant decrease in shoots (34%) and roots (66%) of plants exposed only to GLY; once again, the exogenous application of NO increased APX activity, re-establishing its values to those found in the CTL (Figure 4a,c). Regarding CAT, GLY led to a significant inhibition of its activity in both shoots (53%) and roots (63%), in comparison with the CTL. However, in response to the co-treatment, these negative effects were efficiently counteracted, since no differences were recorded between GLY + NO and CTL plants in shoots and an even higher catalytic activity (1.2-fold increase over the CTL) was found in roots (Figure 4b,d). 

### 2.6. Bioaccumulation of GLY

As can be observed in Figure 5, GLY was only detected in roots of tomato plants exposed to the herbicide, regardless of the co-treatment with NO. Actually, results show that the application of SNP enhanced the root uptake of GLY, with a significant increase of around 33% in comparison with plants grown in the presence of GLY alone. AMPA was not detected in neither roots nor shoots (data not shown).

### 2.7. Productivity-Related Traits

The appearance of the first flower buds occurred upon around 40 days of growth, independent of the presence of GLY in the substrate (data not shown). However, as shown in Table 3, the total number of produced flowers was significantly diminished (51%) by the herbicide, when compared to the CTL. As expected, this reduction in the number of flowers also translated into a decreased fruit set (<55%), whose development was delayed by one week. However, the foliar application of NO prevented some of these effects, as no differences from the CTL were observed for the total number of flowers. Yet, concerning average fruit production, NO was unable to counteract GLY-mediated effects (Table 3), showing values 46% lower than the CTL. Lastly, although no statistical relevance was achieved for the average fresh mass of fruits, a clear tendency can be observed, in which plants exposed to the herbicide alone tend to produce smaller tomatoes in terms of fresh mass (Table 3).

### 2.8. Principal Component Analysis (PCA)

In order to determine how all analysed variables explained the differences among experimental groups, PCA was performed (Figure 6). Results showed that the first component accounted for 43 and 53% of variance in shoots and roots, respectively, and the second for 19% in both organs. Moreover, as can be seen, for roots, CTL and NO plants were clearly grouped together (first quadrant), suggesting that NO alone did not majorly change the growth and physiological status of the plants. In shoots, however, CTL and NO plants were located in distinct quadrants, namely in the second (CTL) and in the third (NO). On the other side, plants exposed to GLY alone were distinctly separated from all other experimental groups, with sample scores being found in the first and second quadrants in shoots and roots, respectively. According to the figure, the parameters that most contributed for this behaviour were the accumulation of proline and GSH, along with ROS overproduction. When plants were grown in the presence of the herbicide, but treated by foliar spraying with NO, an evident effect was also noticed, as this group remained distant from GLY, but closer to the CTL and NO treatments, being the sample scores located in the first/second and third/fourth quadrants in shoots and roots, respectively. 

## 3. Discussion

Given the practical and economical relevance of GLY-based herbicides, more than understanding its non-target phytotoxicity, it is also of particular interest to develop new ecofriendly ways to mitigate its risks to agroecosystems and, in particular, to economically important crops. Yet, work focusing on the implementation of mitigation strategies are still in the beginning. By applying a set of ecophysiological and biochemical endpoints, we show that the foliar application of SNP, a NO donor, can boost *S. lycopersicum*’s tolerance to GLY-contaminated soils (10 mg kg^−1^), improving plant growth by actively controlling the cell redox hub.

### 3.1. GLY Disrupted Tomato Plants’ Growth, but NO Partially Reduced Its Macroscopic Phytotoxicity

Here, it was hypothesized that the exogenous application of NO could protect tomato plants from GLY-induced phytotoxicity. In fact, and in accordance with the before mentioned studies, our results suggest that NO neutralizes, at least to some extent, the negative effects caused by GLY contamination, as shown by a less pronounced growth inhibition in comparison to the CTL. The registered growth inhibition of plants grown in GLY contaminated soil, largely reported by several authors in different plant models [8,9,24,27,28,29], can result from the ability of GLY to decrease the levels of endogenous indole-3-acetic acid (IAA), consequently perturbing cell enlargement and root nodulation [3]. In addition, it can be a consequence of its influence on the synthesis of NR and/or nitrate availability, causing a reduction of the enzyme’s activity [24,28,30,31], as it was reported in roots. 

Aligned with this, data from bioaccumulation studies showed tomato plants were capable of absorbing GLY from the soil solution, and that roots were the preferential organ for GLY storage in plant cells, independent of the NO co-exposure. Despite several studies having detected GLY in the aerial parts of plants grown under herbicide exposure [29], our data strongly suggest a very limited rate of GLY translocation and/or an efficient detoxification mechanism of GLY. Unexpectedly, when SNP was foliar applied to GLY-exposed tomato plants, endogenous levels of the herbicide were increased in roots. Although no report is available concerning NO-mediated effects on GLY uptake and partition in plant tissues, a study aimed at evaluating the phytoremediation potential of *Pistia stratiotes* L. to atrazine (150 μg L^−1^) showed that NO supplementation, via SNP (0.05 mg L^−1^), contributed for a lower phytotoxicity but enhanced the bioaccumulation of this compound [32]. Thus, it appears that NO ameliorative features are most likely related to its function as a signalling molecule, capable of inducing coordinated crosstalk of distinct metabolic chains, rather than inhibiting herbicide uptake and accumulation.

### 3.2. GLY Disrupted the Cellular Redox State, but NO Managed to Keep ROS under Control

Despite being a RNS, the exogenous application of NO to plants exposed to a wide variety of abiotic stresses has been found to prevent the occurrence of oxidative stress [15]. Corroborating the data obtained for biometric analysis, we show that foliar treatment with NO of GLY-exposed plants results in better ROS management, as evidenced by generally reduced levels of O_2_^•−^ (in shoots and roots) and H_2_O_2_ (in roots), when compared to plants only exposed to GLY. Indeed, increased ROS accumulation in response to GLY exposure has been largely documented in different plant species (reviewed by Gomes et al. [3]). Despite the maintenance of H_2_O_2_ levels in the shoots, the MDA content, which reflects the degree of LP in the cellular membranes, was significantly increased upon GLY single treatment, revealing the occurrence of oxidative damage in the aerial parts of tomato plants. This finding, paired with the enhanced accumulation of O_2_^•−^, suggests that downstream-formed ROS can be mediating the occurrence of LP. Although O_2_^•−^ radicals are described as moderate oxidizing agents and cannot be easily diffused through cellular and organelle membranes, evidence suggest that excess of this ROS can indirectly induce substantial oxidative damage by giving rise to more powerful oxidant agents, including the hydroxyl radical (OH^.^) and the hydroperoxyl (HO_2_^.^, a very reactive and stable compound), both able to cross biological membranes and involved in the peroxidation of membrane phospholipids [33,34,35]. 

Due to its lipophilic features, NO can interact with O_2_^•−^ ions, leading to the subsequent formation of peroxynitrite (ONOO), a less toxic compound, thus limiting the downstream production of other ROS capable of inducing LP. Moreover, as reviewed by Arora et al. [22], the reaction between NO and superoxide radicals is far faster than the action of O_2_^•−^-degrading enzyme SOD. In accordance, the increased levels of this ROS in response to GLY were restored back to CTL levels upon co-exposure to NO, and actually decreased to lower values in the shoots. Furthermore, the NO co-treatment even promoted a reduction of H_2_O_2_ content in roots of GLY-exposed plants. In fact, two recent studies by Vieira et al. [32] and Singh et al. [24] have shown that SNP application (0.05 mg L^−1^ (168 µM) and 250 µM) lead to decreased ROS content in *Pista stratiotes* treated with atrazine and *P. sativum* exposed to 0.25 mM GLY, respectively. 

Following the same trend observed for ROS, in the shoots, where the herbicide caused a higher proportion of lipid peroxides, the treatment with NO restored MDA values back to those found in the CTL group. The positive role of NO in LP prevention is most likely related to its ability to act as an antioxidant agent, breaking the reactive chains involved in the LP process [23], which involves activation, propagation and termination steps [34]. In a work conducted with soybean (*Glycine max* L.) plantlets, Ferreira et al. [36] demonstrated that lactofen (0.7 L ha^−1^) boosted the production of lipoperoxides, suggesting the occurrence of LP, but the co-application of SNP (50, 100 and 200 µM SNP; two foliar sprays with a 24 h interval) managed to revert this effect, reducing the accumulation of these subproducts. Curiously, in the roots, data suggested that GLY was not inducing major oxidative damages since MDA levels were diminished; however, it should be stressed out that the decrease of MDA levels does not necessarily equal redox homeostasis. In fact, it is known that ROS, especially OH^.^, which is formed by the Haber–Weiss reaction via O_2_^•−^, H_2_O_2_ and transition metals (e.g., copper–Cu), are dangerous for all kinds of biomolecules, namely proteins and nucleic acids [34]. Accordingly, when looking to the protein content of roots, a major reduction was found in response to GLY. Moreover, plants simultaneously treated with GLY and NO did not present any significant differences from the CTL in what concerns MDA and total protein content, indicating the re-establishment of homeostasis-promoting conditions. 

### 3.3. Antioxidant Metabolites Are Not Directly Related to NO-Mediated Restoration of the Redox Balance Disrupted by GLY

According to the data of the current study, a decrease in the antioxidant capacity of plants subjected to GLY was perceived, with NO treatment not being able to neutralize this effect. Thus, it appears that the non-enzymatic antioxidant system is not actively involved in the alleviation of GLY-induced stress by NO, although a more detailed approach was followed in order to pinpoint the specific response and interaction of different non-enzymatic antioxidants. Due to the nature of phenols biosynthetic process, i.e., the shikimate pathway—the main target of GLY toxicity—it is not surprising that total phenol content was diminished when plants were exposed to this herbicide. In fact, GLY-mediated reduction in phenolic compounds has already been documented by some authors [37,38]. Curiously, we report that NO application, with or without GLY co-presence, also led to a decrease in plant phenols and flavonoids in shoots, in contrast to what has been found in the literature. Proline and GSH, two important players in the non-enzymatic component of the antioxidant system, have already been shown to be strongly induced in plants exposed to GLY [8,9,27], in accordance with what is herein reported for both analysed organs. However, despite the observed increases in GSH and proline levels, ROS accumulation took place in shoots and roots, revealing that the modulation of their redox state is not able to limit the toxic effects of GLY on tomato’s oxidative status. In opposition, plants exposed to GLY but simultaneously treated with NO presented proline and GSH levels similar to the CTL, this being accompanied by a better growth performance. The reduction of free GSH levels in GLY + NO treated plants, in comparison to GLY plants, can be related to GSH ability to chemically react with different ROS or its use as a substrate in the enzymatic regeneration of AsA. Indeed, it is known that GSH can eliminate ROS excess, such as O_2_^•−^, which were clearly reduced in plants co-exposed to GLY and NO. Moreover, and confirming our previous hypothesis raised in Fernandes et al. [39], it becomes apparent that proline may not be a key player in modulating tolerance to this herbicide, and the reduction of its levels in NO-treated plants can be a consequence of stress alleviation through other mechanisms. Additionally, it is also important to highlight that the exacerbated accumulation of proline in GLY-exposed plants could also have prevented a disbalance in the cellular osmotic potential, as the water status of plants was not altered by the herbicide, as previously reported in tomato plants exposed to 10 mg kg^−1^ GLY [25]. Following the trend recorded for proline and GSH, the accumulation of AsA in response to the co-treatment with NO was somewhat distinct from that of plants exposed to the herbicide alone. Indeed, the higher AsA/DHA ratio found in shoots and roots of co-treated plants suggest that, upon application of NO, AsA is being actively recruited by APX, as there appears to be an upregulation of the AsA–GSH cycle, possibly pointing towards a tightly regulated enzymatic regeneration mechanism focused on maintaining a sufficient AsA pool to fulfil the antioxidant needs of *S. lycopersicum* plants. In fact, the stimulation of AsA production when plants were treated with NO during the exposure to different contaminants, such as metals [21,40,41] and herbicides [42,43], has been extensively reported

### 3.4. NO-Mediated Alleviation of GLY Phytotoxicity Involves the Upregulation of the Main Antioxidant Enzymes

For both organs, there was a striking pattern that shows GLY acting as a powerful inhibitor of enzyme activity, as SOD, CAT and APX action were severely hindered when *S. lycopersicum* plants were grown in GLY-contaminated soils. Up to now, distinct findings have been published concerning the effects of GLY on the performance of the plant antioxidant system [3]. Here, the inhibition of SOD is tightly related to the observed increase in O_2_^•−^ in both shoots and roots of tomato plants grown in GLY-treated soils. However, CAT- and APX-reduced activity did not result in an overaccumulation of H_2_O_2_, reinforcing the idea that tomato plants depend primarily on their non-enzymatic defences to deal with GLY toxic levels intracellularly. Despite the overall inhibition of the main antioxidant enzymes in response to GLY, when exogenous NO was supplied, all enzymes (SOD, CAT and APX) were restored, or even increased. Accordingly, the upregulation of several enzymatic antioxidant players by the exogenous application of NO has been reported by different authors and studies [21,40,42,43,44,45], including in plants exposed to metals such as cadmium [41] and copper [46], and herbicides, for example, atrazine, glufosinate [43] and even GLY [24]. In this sense, it is possible to hypothesize that NO-induced redox balance of GLY-treated plants is tightly related to a stimulation or a restoration of the enzymatic component of the antioxidant system. Moreover, taking into account the possible impact of GLY on the activity of metalloenzymes, by chelating their important co-factors, it is possible that not only NO could be acting by enhancing the efficiency of the enzymatic antioxidant system, but also by stimulating GLY detoxification pathways, protecting the protein structure of SOD, CAT and APX. Nonetheless, to further prove this hypothesis, subsequent studies to be done should use native polyacrylamide gel electrophoresis to disclose the activity of specific isoenzymes [9,47,48]. This is especially important for SOD, since its various isoforms differ in their metallic co-factors, which are known to be affected by GLY [3].

### 3.5. Detoxification Pathways Impaired by GLY Are Stimulated by the Exogenous Application of NO

Throughout evolution, plants have developed an efficient xenobiotic detoxification system [49,50], which involves the conjugation of the transformed compound to GSH or glucose, through the action of GST or glucosyl-transferases (EC 2.4.-.-), respectively. This process depends on the original characteristics of the xenobiotic, but GST-mediated GLY conjugation has already been suggested by several authors [51,52]. Curiously, our results show an opposite effect, in which plants grown under GLY contamination had a marked decrease in GST activity in both organs. Thus, it appears that, under GLY exposure, roots of tomato plants failed to employ efficient detoxification systems. From what it appears, it is possible that GLY can be interfering with the structure and activity of GST, which results in a poor detoxification process and increased phytotoxic potential, reflected by the severe impairment of plant growth when exposed to this herbicide. A similar finding was also reported in *Lemna minor* L. exposed to diclofenac [53]. In shoots, surprisingly, activity levels of GST were also decreased in GLY-exposed plants, even though GLY was not detected in this organ. However, following the same trend recorded for the other antioxidant enzymes, this finding can reflect the harsh oxidative status that shoots underwent. In fact, it is known that GST can be highly inactivated by ROS, including O_2_^•−^ [54]. In response to the co-application of NO, GST activity in roots was restored back to CTL levels, even though a higher bioaccumulation of GLY has been found. This improved, or at least re-established detoxification process made use of the existing GSH pool, to conjugate this thiol with GLY, forming fewer toxic metabolites. Through this process, NO-treated *S. lycopersicum* plants seemed to have been able to reduce GLY toxicity and to improve their growth and performance under these adverse conditions. In fact, increased GST activity in plants treated with this molecule has been reported after exposure to paraquat [42] and several metals [15], which share a common detoxification mechanism with xenobiotics.

### 3.6. GLY-Mediated Effects on Crop Productivity Are Partially Prevented by the Co-Application of NO

In addition to affecting plant growth and biomass production, soil residues of GLY (10 mg kg^−1^) have also resulted in a declined number of flowers and fruits, impacting the fresh mass of the produced tomatoes. Accordingly, a recent study conducted by Strandberg et al. [55] concluded that, while GLY spray-drift had no effect on flowering time, it adversely affected the cumulative number of flowers of native non-target species (*Trifolium pratense* L. and *Lotus corniculatus* L.). Yet, the assessment of *Brassica* sp. reproductive responses to a GLY-based herbicide (Roundup^®^) pointed towards the occurrence of major changes in the flowering time and reproductive function, especially male gametophytes [56]. Actually, it is known that even GLY-resistant crops can experience substantial changes in their reproductive traits, with major consequences on fruit production [57]. Aligned with this significant reduction in the number of flowers, fruits from GLY-exposed plants were fewer and smaller than those produced from CTL plants, revealing that soil residues of this herbicide also negatively impact the overall productivity of the plant [58]. Up to now, studies dealing with the possible effects of GLY soil contamination on fruit production of non-target crops are scarce [58,59], this being one of the first records exploring this issue. Based on the data herein collected, one can hypothesize that GLY-mediated impacts on tomato plants productivity mostly arise as a consequence of the physiological disturbances induced by the herbicide, rather than the effects of GLY itself, since no bioaccumulation was found in shoots. In accordance to our hypothesis, recent findings suggest that composts obtained from earthworms exposed to GLY can disrupt tomato development and ability to flower [60], especially due to GLY-mediated chelation of essential nutrients, which become unavailable for plant growth.

The overall positive effects of NO against GLY-mediated toxicity on the growth and antioxidant response of tomato plants were also evident in the flowering process. As reviewed by Sun et al. [61], NO was already proved to benefit plant reproductive traits, inducing the expression of several flowering-related genes. Moreover, although the total number of produced tomatoes was still lower than that of the CTL, fruits’ average fresh mass was improved and remained identical to unexposed plants. Indeed, NO application has been found to modulate fruit quality features, contributing for a better firmness and to delay fruit ripening, by inhibiting ethylene biosynthesis [61].

## 4. Materials and Methods

### 4.1. Chemicals and Test Substrate

Roundup^®^ UltraMax (Monsanto Europe, S.A., Belgium), whose active compound is GLY (360 g GLY L^−1^, potassium salt), was acquired from a local supplier. This formulation was diluted in deionized water to prepare a stock solution of 1 g GLY L^−1^, later used for obtaining the required amount of GLY to be added to the soil (10 mg GLY kg^−1^). Sodium nitroprusside (SNP; Sigma-Aldrich^®^), used as NO donor, was diluted in deionized water to obtain a solution of 200 µM. An artificial soil (pH 6.0 ± 0.5), composed by sphagnum peat, quartz sand (<2 mm) and kaolin clay (5:72.5:22.5), prepared according to OECD (Organisation for Economic Co-operation and Development) standards [62], was used in this study.

### 4.2. Plant Material, Plant Growth Conditions and Experimental Design

Seeds of *S. lycopersicum* cv. Micro-Tom were surface disinfected for 7 min with 70% (*v*/*v*) ethanol, followed by 5 min with 20% (*v*/*v*) commercial bleach (5% active chloride) mixed with 0.02% (*w*/*v*) Tween-20, and then rinsed several times with deionized water. Afterwards, seeds were germinated in Petri dishes (10 cm diameter) with 0.5× Murashige and Skoog (MS) medium [63] solidified with 0.625% (*w*/*v*) agar, in a growth chamber (temperature: 25 °C; photoperiod: 16 h light/8 h dark; photosynthetic active radiation (PAR): 60 mmol m^−2^ s^−1^). After 10 days, seedlings were selected and transferred to plastic pots (5 seedlings per pot) filled with 200 g_dry_ OECD soil, which was moistened with deionized water to obtain 40% of its maximum water holding capacity (WHC_max_), previously determined according to ISO [64]. To acquire a homogenous mixture, the soil was manually mixed. For GLY-contaminated soils, the amount of herbicide needed to obtain a 10 mg kg^−1^ concentration was taken from the stock solution of 1 g L^−1^. The selection of the GLY concentration was based on our previous work, the recommended dosage used in agriculture and studies on soil contamination by GLY [8]. The first watering was done with a half-strength Hoagland solution (pH 5.8) [65] in order to avoid nutrient deficiency. Deionized water was then added as needed to maintain soil moisture.

With the purpose of understanding the potential ameliorative role of NO against GLY-induced toxicity, the following experimental groups were considered: CTL—control plants, grown in the absence of GLY and foliar sprayed with dH_2_O once a week (negative control); NO—plants grown in the absence of GLY and foliar sprayed with SNP (200 μM) once a week; GLY—plants grown in the presence of GLY (10 mg kg^−1^) (positive control); GLY + NO—plants grown in the presence of GLY and weekly sprayed with SNP.

For each experimental group, 12 experimental replicates were prepared (8 pots each one with 5 seedlings). After 28 days of growth in a growth chamber (PAR: 120 μmol m^−2^ s^−1^; photoperiod 16 h light/8 h dark; temperature: 25 °C), plants were harvested and divided into shoots and roots. Part of the biological material (4 replicates) was used immediately to evaluate the biometric parameters, and to determine the levels of superoxide anion (O_2_^•−^), while the plant material from other 4 replicates was frozen with liquid nitrogen and kept at −80 °C for further analyses. The remaining set of plants (*n* = 4) were grown until maturity for the estimation of productivity traits (number of flowers, and number and fresh mass of fruits). For all biometric, biochemical and productivity-related endpoints evaluated, aliquots from at least three experimental replicates were used (n ≥ 3).

### 4.3. Biometric and Productivity-Related Analysis

After the growth period (28 days), the roots were washed with tap and deionized water, and their length was measured. Following the separation of roots and shoots, the fresh biomass of both organs (roots and shoots) was determined using a precision balance (KERN^©^ EWJ 300-3; KERN & SOHN GmbH, Balingen, Germany). Concerning productivity-related traits, a set of plants was left until maturity, in order to monitor the total number of flowers and fruits, and the total fresh mass of produced tomatoes.

### 4.4. Total Protein Content and NR (EC 1.7.1.1) Activity

Total soluble protein and NR from shoots and roots were extracted in frozen aliquots (ca. 200 mg) by homogenizing samples in an appropriate extraction buffer (50 mM HEPES-KOH (pH 7.8), 1 mM phenylmethylsulfonyl fluoride (PMSF) and 10 mM magnesium chloride (MgCl_2_)) under cold conditions. After centrifugation (25 min; 15,000× *g*; 4 °C), the supernatants (SN) were collected and used for protein quantification [66] and for NR activity measurements. The determination of NR activity was performed through enzyme kinetics in accordance to Kaiser and Brendle-Behnisch [67]. The proposed procedure was scaled-down to an UV microplate and the assays were performed in a microplate reader (Thermo Scientific™ Multiskan™ GO Microplate Reader). Activity levels were expressed as mmol min^−1^ mg^−1^ of protein, using the NADH extinction coefficient (6.22 mM^−1^ cm^−1^).

### 4.5. Biomarkers of Oxidative Stress

#### 4.5.1. O_2_^•−^ and H_2_O_2_

The levels of O_2_^•−^ were quantified according to the method described by Gajewska and Skłodowska [68], using fresh plant material of roots and shoots (200 mg). After a 1 h reaction at dark conditions in a reaction mixture (2 mL), containing nitroblue tetrazolium (NBT) and sodium azide (NaN_3_), an incubation period of 15 min at 85 °C was followed. At the end, the absorbance (Abs) of the obtained solution was recorded at 580 nm and O_2_^•−^ levels were expressed in Abs_580nm_ h^−1^ g^−1^ fresh mass (f.m.). The quantification of H_2_O_2_ levels was performed in frozen samples, following the spectrophotometric assay of Alexieva et al. [69], which is based on the reaction between H_2_O_2_ and potassium iodide, forming a yellowish complex, measurable at 390 nm. Its content was determined through a standard curve, using known concentrations of H_2_O_2_ and later expressed in nmol g^−1^ f.m.

#### 4.5.2. LP

LP was estimated by the evaluation of malondialdehyde (MDA) content, via spectrophotometry, following the procedure described by Heath and Packer [70]. Abs was recorded at 532 and 600 nm. The difference between Abs_532_ and Abs_600_ was calculated to eliminate non-specific turbidity. Considering the ε of 155 mM^−1^ cm^−1^, MDA content was determined and expressed in nmol MDA g^−1^ f.m.

### 4.6. Evaluation of Antioxidant Metabolites

#### 4.6.1. Quantification of AsA, GSH and Proline

The quantification of total, reduced and oxidized (dehydroascorbate; DHA) AsA was accomplished by following the procedure proposed by Gillespie and Ainsworth [71]. This method allows the quantification of reduced AsA, through the 2,2′-bipyridyl method. Total AsA was determined via the same method, but with the addition of dithiothreitol (DTT) to reduce DHA. After 1 h at 37 °C, the Abs of each sample was read at 525 nm and DHA content was determined by the difference between total and reduced AsA levels. Results were expressed as µmol AsA g^−1^ f.m. by comparison with a standard curve prepared with stock solutions of AsA.

To determine free GSH levels, a spectrophotometric assay adapted from a commercial kit was followed as described by Soares et al. [8]. After the extraction procedure (3% (*w*/*v*) sulphosalicylic acid), samples were centrifugated at 4 °C and the SN was mixed with 5,5-dithio-bis-(2-nitrobenzoic acid) (DTNB; 1.5 mg mL^−1^). After 10 min, the Abs at 412 nm was registered and GSH levels were expressed as nmol GSH g^−1^ f.m. with the aid of a calibration curve prepared with known GSH concentrations.

Proline levels were estimated through a colorimetric ninhydrin-based assay described by Bates et al. [72]. Samples from shoots and roots (200 mg) were homogenised with 3% (*m*/*v*) sulphosalicylic acid and centrifuged (500× *g*; 10 min). Then, an incubation of 1 h at 95 °C was performed, in which the SN reacted with ninhydrin in an acidic medium. In the end, the Abs was recorded at 520 nm and the results were expressed in μg g^−1^ f.m., using known concentrations of proline to establish a standard curve.

#### 4.6.2. Determination of TPC, Total Flavonoids and TAC

The estimation of TPC, flavonoids and TAC was achieved by adapting the procedure described by Zafar et al. [73]. For that, frozen samples were homogenised, on ice, with 80% (*v*/*v*) methanol and centrifuged (10 min; 2500× *g*). Regarding TPC, the SN reacted with Folin–Ciocalteu reagent and, after 5 min at room temperature (RT), 7.5% (*w*/*v*) sodium carbonate (Na_2_CO_3_) was added. Samples were then incubated for 1 h in dark conditions at RT. Lastly, the Abs was recorded at 725 nm and results were expressed in mg gallic acid equivalents g^−1^ f.m., using a calibration curve prepared with standard solutions of gallic acid. Concerning total flavonoids, the methanolic extracts were incubated with 10% (*m*/*v*) aluminium chloride (AlCl_3_) and 1 M potassium acetate (CH_3_CO_2_K), for 30 min at RT. Afterwards, the Abs of each sample was read at 415 nm and the levels extrapolated from a linear calibration curve, prepared with quercetin standards. For TAC, the methanolic extracts were properly diluted (1:3 in methanol) and added to a reagent solution containing 0.6 M sulphuric acid, 4 mM ammonium molybdate and 28 mM sodium phosphate, followed by incubation for 90 min at 95 °C. Afterwards, the Abs was recorded at 695 nm. Results were expressed in mg AsA equivalents g^−1^ f.m. (TAC), using a calibration curve prepared with standard solutions of AsA.

### 4.7. Extraction of Antioxidant Enzymes

The main ROS-scavenging enzymes were extracted in accordance with the method described by Soares et al. [8], using frozen aliquots of shoots (200 mg in 1.5 mL of extraction buffer) and roots (200 mg in 1.2 mL of extraction buffer). Upon centrifugation (16,000× *g*; 25 min; 4 °C), SN was collected and transferred to new tubes for enzyme activity assessment and soluble protein quantification [66].

### 4.8. Spectrophotometric Activity Quantification of SOD (EC 1.15.1.1), CAT (EC.1.11.1.6), APX (EC.1.11.1.11) and GST (EC2.5.1.18)

Total activity of SOD was estimated through spectrophotometry (Abs at 560 nm), based on the inhibition of the photochemical reduction of nitro blue tetrazolium (NBT), according to Donahue et al. [74]. Results were expressed as units of SOD mg^−1^ of protein, in which one unit of SOD corresponds to the amount of enzyme required to cause 50% inhibition of the NBT photoreduction rate.

GST activity was estimated following the procedure described by Teixeira et al. [75], measuring the increase of the GSH-2,4-dinitrochlorobenzene (CDNB) complex at 340 nm. Results were expressed in nmol conjugated CDNB min^−1^ mg^−1^ of protein, using an ε of 9.6 mM^−1^ cm^−1^.

Both CAT and APX activity were determined by enzyme kinetics (Abs at 240 and 290 nm, respectively), as described by Aebi [76] and Nakano and Asada [77], following the degradation of H_2_O_2_ (ε_240 nm_ = 39.4 M^−1^ cm^−1^) and AsA (ε_290 nm_ = 2.8 mM^−1^ cm^−1^), respectively, and expressed as µmol H_2_O_2_ min^−1^ mg^−1^ of protein or µmol AsA min^−1^ mg^−1^ of protein, respectively. In either case, the reaction was started by the addition of H_2_O_2_. The original protocols were adapted to UV microplates, based on the optimization of Murshed et al. [78].

### 4.9. Analytical Quantification of GLY and AMPA

The extraction of GLY from roots and shoots of tomato samples was performed as described elsewhere (AOAC official method 2000.05) and fully detailed by Soares et al. (submitted). All subsequent analyses were performed based on Pinto et al. [79], with some modifications: 1 mL of the extract (SN) was diluted with 1 mL of internal standard (200 μg L^−1^ of glyphosate 1,2-13C2 15N and 200 μg L^−1^ of 13C,15N-AMPA), and then added to 120 µL of 1% (*m*/*v*) NH_4_OH solution and 120 µL of FMOC-Cl (12,000 mg L^−1^ in acetone). Afterwards, samples were vortexed and incubated for 30 min at RT. To stop the reaction, 10 μL of 6 M HCl were added. The samples derived were filtered through a 0.45 μm PTFE filters into LC vials. GLY and AMPA were determined by liquid chromatography with tandem mass spectrometry (LC–MS/MS) using the internal standard method.

The LC–MS/MS system included a Waters 2695 XE separation module (Milford, MA) interfaced with a triple quadrupole mass spectrometer (Quattro micro™ API triple quadrupole, Waters Micro-mass, Manchester, UK). The LC separation was performed using a Kinetex^®^ EVO C18 core-shell column (2.6 µm; 100 × 2.1 mm; flow rate of 225 µL min^−1^). A binary gradient was used: solvent A (10 mM ammonium bicarbonate) and solvent B (methanol). The percentage of organic modifier (B) was gradually modified as follows: 0–0.5 min, 5%; 0.5–5.5 min, 90%; 5.5–6.5 min, 90%; 6.5–6.7 min, 5%; 6.7–14 min, 5%. A total of 20 µL of each sample was injected and the analyses were performed at 40 °C. The mass spectrometry parameters were as follows: ion mode, positive; capillary voltage, 3.00 kV; source temperature, 130 °C; desolvation temperature, 450 °C; desolvation gas flow, 600 L/h; and multiplier, 650 V. High purity nitrogen (>99.999%) and argon (>99.999%) were used as the cone and collision gases, respectively. The precursor and product ions, along with the cone voltage and collision energy for each GLY-FMOC, AMPA-FMOC and ILIS-FMOC, were measured by flow injection analysis and the MRM transitions, cone voltages and collision energies are listed in Appendix A. Data acquisition was performed by the MassLynx V4.1 software. Results were expressed as µg g^−1^ d.m.

### 4.10. Statistical Analysis

All biometric and biochemical analysis were performed considering at least three experimental replicates (n ≥ 3). Results were expressed as mean ± standard deviation (SD). After checking data homogeneity (Brown–Forsythe test), one-way ANOVA was performed in conjunction with Tukey’s post hoc test, assuming 0.05 as a significance level (p). All statistical analyses were performed in GraphPad Prism® 8 (San Diego, CA, USA). In order to execute a principal component analyses (PCA), all evaluated parameters (biometric and biochemical) from each experimental group were plotted to investigate the main factors behind the observed differences. These procedures were performed in the software XLSTAT 2021.2.2 (http://www.xlstat.com, accessed on 3 August 2021, Addinsoft USA, New York, NY, USA). The statistical data reporting the results of ANOVA analyses can be found in Appendix A of the Appendix A.

## 5. Conclusions

As can be seen in the PCA (Figure 6), tomato plants responded differentially to the presence of GLY in the soil, undergoing a state of oxidative stress and impaired growth, especially in the non-green tissues. However, the foliar application of NO successfully improved tomato plant growth and development, with a clear separation from plants exposed to the herbicide alone. According to the biochemical data, this NO-mediated protection was mainly due to its features as radical scavenger and stimulator of antioxidant mechanisms, contributing for the restoration of the cellular redox status and, consequently, leading to an increased growth potential under herbicide co-exposure. Moreover, the phytoprotective role of NO was also evident when reproductive and productivity traits were evaluated, since the number of flowers and fresh mass of produced tomatoes was increased in comparison with plants only exposed to the herbicide. Overall, this is the first study exploring the benefits of NO supplementation for non-target crops growing in GLY contaminated soils using an environmentally relevant approach, covering growth- and productivity-related endpoints. In the future, in order to concretely assess if the foliar application of NO, through its donor SNP, can represent an effective tool for plant stress management, it would be of great interest i) to test other modes-of-application and concentrations of this molecule throughout the plant’s life cycle (vegetative and reproductive phases) and ii) to study the influence of GLY and NO co-exposure on tomato nutritional and antioxidant profile to ensure food safety, quality and security.

## Figures and Tables

**Figure 1 plants-10-01862-f001:**
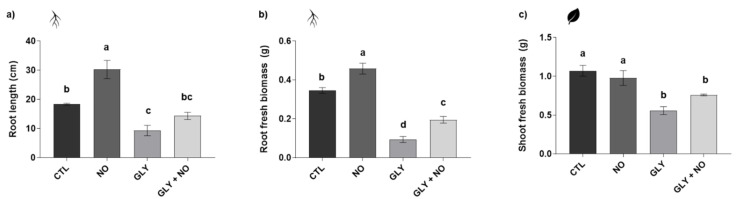
Growth traits of *Solanum lycopersicum* L. cv. Micro-Tom grown for 28 days in an artificial soil contaminated by GLY (10 mg kg^−1^) and/or foliar sprayed with SNP (200 µM): (**a**) root length; (**b**) root fresh biomass; (**c**) shoot fresh biomass. CTL—control plants, grown in the absence of GLY and foliar sprayed with dH_2_O once a week (black); NO—plants grown in the absence of GLY and foliar sprayed with SNP once a week (dark grey); GLY—plants grown in the presence of GLY (grey); GLY + NO—plants grown in the presence of GLY and weekly sprayed with SNP (light grey). Results are presented as mean ± standard deviation (SD) and result from the evaluation of at least three experimental replicates (*n* ≥ 3). Different letters above bars indicate significant differences between groups (CTL, NO, GLY and GLY + NO) at *p* ≤ 0.05, according to the one-way ANOVA followed by Tukey’s post hoc test.

**Figure 2 plants-10-01862-f002:**
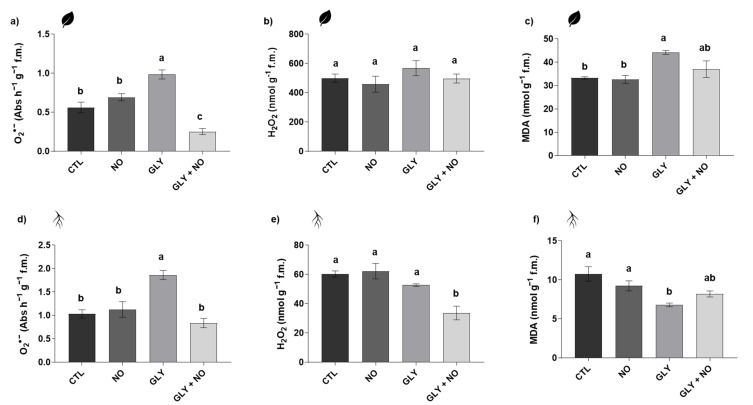
Redox status of *Solanum lycopersicum* L. cv. Micro-Tom grown for 28 days in an artificial soil contaminated by GLY (10 mg kg^−1^) and/or foliar sprayed with SNP (200 µM): (**a**,**d**) superoxide anion (O_2_^•−^) content; (**b**,**e**) hydrogen peroxide (H_2_O_2_) content; (**c**,**f**) malondialdehyde (MDA) levels. CTL—control plants, grown in the absence of GLY and foliar sprayed with dH_2_O once a week (black); NO—plants grown in the absence of GLY and foliar sprayed with SNP once a week (dark grey); GLY—plants grown in the presence of GLY (grey); GLY + NO—plants grown in the presence of GLY and weekly sprayed with SNP (light grey). Results are presented as mean ± standard deviation (SD) and result from the evaluation of at least three experimental replicates (*n* ≥ 3). Different letters above bars indicate significant differences between groups (CTL, NO, GLY and GLY + NO) at *p* ≤ 0.05, according to the one-way ANOVA followed by Tukey’s post hoc test.

**Figure 3 plants-10-01862-f003:**
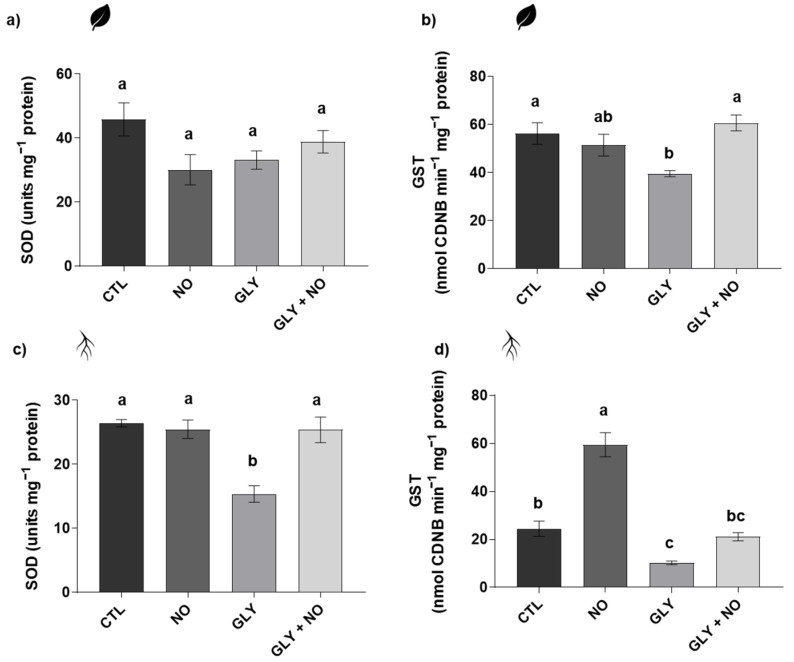
Activity of antioxidant enzymes of *Solanum lycopersicum* L. cv. Micro-Tom grown for 28 days in an artificial soil contaminated by GLY (10 mg kg^−1^) and/or foliar sprayed with SNP (200 µM): (**a**,**c**) superoxide dismutase (SOD) and (**b**,**d**) glutathione-S-transferase (GST). CTL—control plants, grown in the absence of GLY and foliar sprayed with dH_2_O once a week (black); NO—plants grown in the absence of GLY and foliar sprayed with SNP once a week (dark grey); GLY—plants grown in the presence of GLY (grey); GLY + NO—plants grown in the presence of GLY and weekly sprayed with SNP (light grey). Results are presented as mean ± standard deviation (SD) and result from the evaluation of at least three experimental replicates (*n* ≥ 3). Different letters above bars indicate significant differences between groups (CTL, NO, GLY and GLY + NO) at *p* ≤ 0.05, according to the one-way ANOVA followed by Tukey’s post hoc test.

**Figure 4 plants-10-01862-f004:**
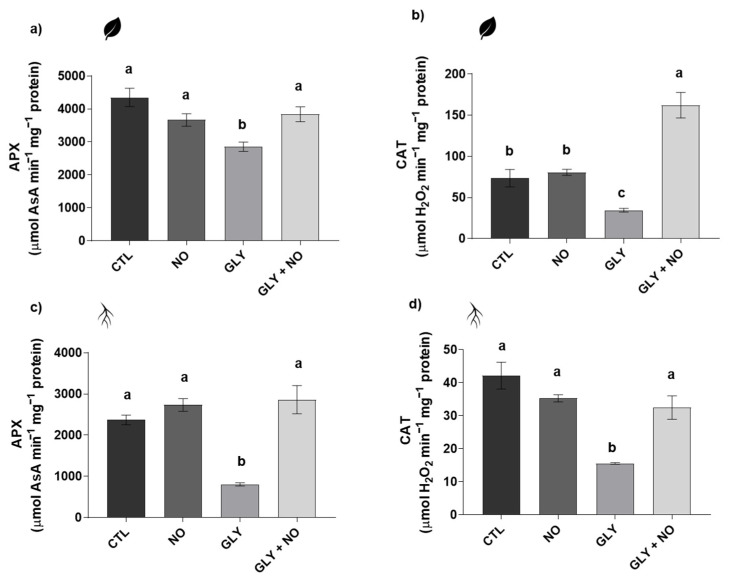
Activity of antioxidant enzymes of *Solanum lycopersicum* L. cv. Micro-Tom grown for 28 days in an artificial soil contaminated by GLY (10 mg kg^−1^) and/or foliar sprayed with SNP (200 µM): (**a**,**c**) ascorbate peroxidase (APX) and (**b**,**d**) catalase (CAT). CTL—control plants, grown in the absence of GLY and foliar sprayed with dH_2_O once a week (black); NO—plants grown in the absence of GLY and foliar sprayed with SNP once a week (dark grey); GLY—plants grown in the presence of GLY (grey); GLY + NO—plants grown in the presence of GLY and weekly sprayed with SNP (light grey). Results are presented as mean ± standard deviation (SD) and result from the evaluation of at least three experimental replicates (*n* ≥ 3). Different letters above bars indicate significant differences between groups (CTL, NO, GLY and GLY + NO) at *p* ≤ 0.05, according to the one-way ANOVA followed by Tukey’s post hoc test.

**Figure 5 plants-10-01862-f005:**
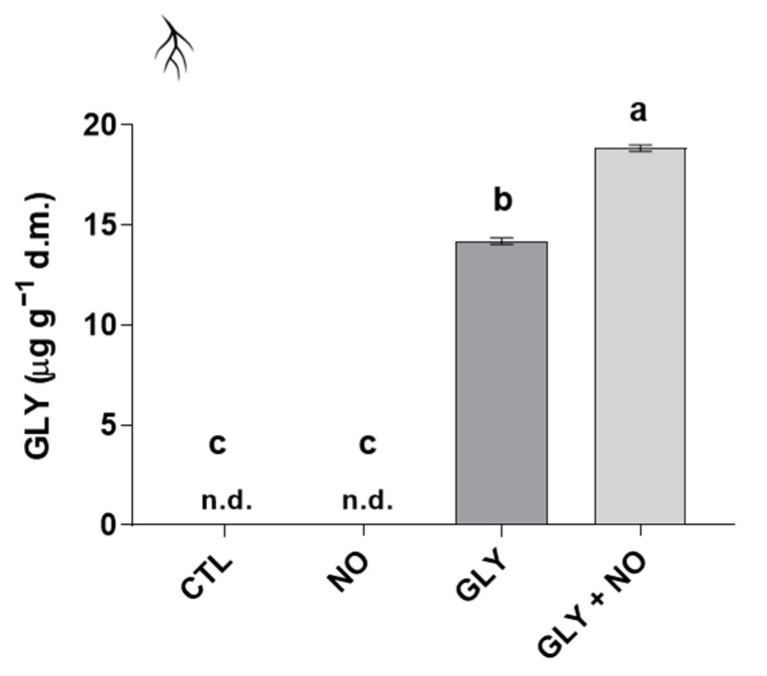
Bioaccumulation of GLY in roots of *Solanum lycopersicum* L. cv. Micro-Tom grown for 28 days in an artificial soil contaminated by GLY (10 mg kg^−1^) and/or foliar sprayed with SNP (200 µM). CTL—control plants, grown in the absence of GLY and foliar sprayed with dH_2_O once a week (black); NO—plants grown in the absence of GLY and foliar sprayed with SNP once a week (dark grey); GLY—plants grown in the presence of GLY (grey); GLY + NO—plants grown in the presence of GLY and weekly sprayed with SNP (light grey). Results are presented as mean ± standard deviation (SD) and result from the evaluation of at least three experimental replicates (*n* ≥ 3). Different letters above bars indicate significant differences between groups (CTL, NO, GLY and GLY + NO) at *p* ≤ 0.05, according to the one-way ANOVA followed by Tukey’s post hoc test; n.d.: non-detected, which means below the detection limit.

**Figure 6 plants-10-01862-f006:**
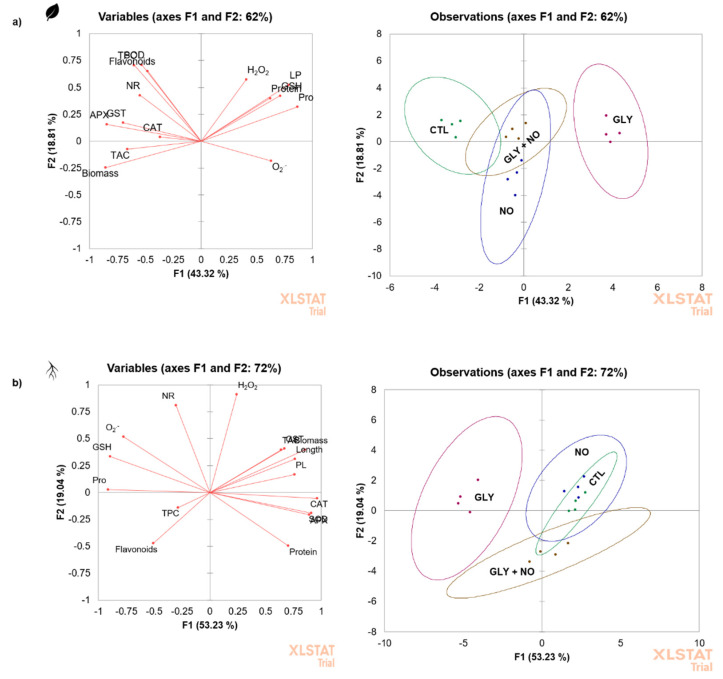
Principal component analysis (PCA) (xx axis—first component, yy axis—second component) of all evaluated endpoints (biometrical and biochemical) in (**a**) shoots and (**b**) roots of *Solanum lycopersicum* L. cv. Micro-Tom grown for 28 days in an artificial soil contaminated by GLY (10 mg kg^−1^) and/or foliar sprayed with SNP (200 µM). CTL—control plants, grown in the absence of GLY and foliar sprayed with dH_2_O once a week (green points); NO—plants grown in the absence of GLY and foliar sprayed with SNP once a week (blue points); GLY—plants grown in the presence of GLY (purple points); GLY + NO—plants grown in the presence of GLY and weekly sprayed with SNP (brown points).

**Table 1 plants-10-01862-t001:** Biochemical parameters (total protein, nitrate reductase (NR) activity, proline, total ascorbate, ascorbate:dehydroascorbate (AsA/DHA) ratio, glutathione (GSH), total antioxidant capacity (TAC), total phenols and flavonoids) of shoots of *Solanum lycopersicum* L. cv. Micro-Tom grown for 28 days in an artificial soil contaminated by GLY (10 mg kg^−1^) and/or foliar sprayed with SNP (200 µM). CTL—control plants, grown in the absence of GLY and foliar sprayed with dH_2_O once a week; NO—plants grown in the absence of GLY and foliar sprayed with SNP once a week; GLY—plants grown in the presence of GLY; GLY + NO—plants grown in the presence of GLY and weekly sprayed with SNP. Results are presented as mean ± standard deviation (SD) and result from the evaluation of at least three experimental replicates (*n* ≥ 3). Different letters indicate significant differences between groups (CTL, NO, GLY and GLY + NO) at *p* ≤ 0.05, according to the one-way ANOVA followed by Tukey’s post hoc test.

Parameter	CTL	NO	GLY	GLY + NO
Total protein (mg g^−1^ f.m.)	3.03 ± 0.20 b	3.47 ± 0.19 ab	3.85 ± 0.14 a	3.68 ± 0.21 ab
NR (mmol NADH min^−1^ mg^−1^ protein)	41.67 ± 3.18 a	26.67 ± 1.76 b	28.33 ± 3.18 b	25 ± 2.88 b
Proline (μg g^−1^ f.m.)	110 ± 8 b	88 ± 11 b	587 ± 87 a	163 ± 26 b
Total ascorbate (µmol g^−1^ f.m.)	1.52 ± 0.09 bc	1.29 ± 0.18 c	2.05 ± 0.18 ab	2.27 ± 0.21 a
AsA/DHA	2.94 ± 0.66 a	0.37 ± 0.10 b	1.47 ± 0.22 a	2.120 ± 0.40 a
GSH (nmol g^−1^ f.m.)	288 ± 21 b	310 ± 20 b	454 ± 9 a	426 ± 20 a
TAC (μg AsA equivalents g^−1^ f.m.)	1067 ± 141 a	928 ± 133 a	717 ± 84 a	895 ± 112 a
Total phenols (μg gallic acid equivalents g^−1^ f.m.)	960 ± 27 a	381 ± 10 c	542 ± 28 bc	652 ± 54 b
Flavonoids (μg quercetin equivalents g^−1^ f.m.)	424 ± 43 a	217 ± 14 b	298 ± 8 b	326 ± 16 ab

f.m.: fresh mass.

**Table 2 plants-10-01862-t002:** Biochemical parameters (total protein, nitrate reductase (NR) activity, proline, total ascorbate, ascorbate:dehydroascorbate (AsA/DHA) ratio, glutathione (GSH), total antioxidant capacity (TAC), total phenols and flavonoids) of roots of *Solanum lycopersicum* L. cv. Micro-Tom grown for 28 days in an artificial soil contaminated by GLY (10 mg kg^−1^) and/or foliar sprayed with SNP (200 µM). CTL—control plants, grown in the absence of GLY and foliar sprayed with dH_2_O once a week; NO—plants grown in the absence of GLY and foliar sprayed with SNP once a week; GLY—plants grown in the presence of GLY; GLY + NO—plants grown in the presence of GLY and weekly sprayed with SNP. Results are presented as mean ± standard deviation (SD) and result from the evaluation of at least three experimental replicates (*n* ≥ 3). Different letters indicate significant differences between groups (CTL, NO, GLY and GLY + NO) at *p* ≤ 0.05, according to the one-way ANOVA followed by Tukey’s post hoc test.

Parameter	CTL	NO	GLY	GLY + NO
Total protein (mg g^−1^ f.m.)	5.09 ± 0.55 a	3.66 ± 0.21 b	2.53 ± 0.10 c	4.91 ± 0.14 a
NR (mmol NADH min^−1^ mg^−1^ protein)	110 ± 12 a	122 ± 11 a	132 ± 15 a	83 ± 10 a
Proline (μg g^−1^ f.m.)	46.44 ± 1.98 b	46.38 ± 3.73 b	95.74 ± 10.68 a	58.17 ± 7.52 b
Total ascorbate (µmol g^−1^ f.m.)	0.40 ± 0.05 b	0.64 ± 0.05 a	0.41 ± 0.04 b	0.38 ± 0.03 b
AsA/DHA	0.77 ± 0.05 bc	0.68 ± 0.12 c	1.02 ± 0.07 ab	1.11 ± 0.07 a
GSH (nmol g^−1^ f.m.)	68.96 ± 1.31 b	65.69 ± 4.67 b	139.1 ± 9.53 a	58.49 ± 2.88 b
TAC (μg AsA equivalents g^−1^ f.m.)	422 ± 35 a	362 ± 14 a	309 ± 21 b	315 ± 20 b
Total phenols (μg gallic acid equivalents g^−1^ f.m.)	243 ± 9 a	273 ± 32 a	280 ± 38 a	273 ± 17 a
Flavonoids (μg quercetin equivalents g^−1^ f.m.)	19.2 ± 0.9 a	21.0 ± 0.2 a	24.1 ± 2.8 a	24.3 ± 4.4 a

f.m.: fresh mass.

**Table 3 plants-10-01862-t003:** Productivity-related characteristics of *Solanum lycopersicum* L. cv. Micro-Tom grown for 28 days in an artificial soil contaminated by GLY (10 mg kg^−1^) and/or foliar sprayed with SNP (200 µM). CTL—control plants, grown in the absence of GLY and foliar sprayed with dH_2_O once a week; NO—plants grown in the absence of GLY and foliar sprayed with SNP once a week; GLY—plants grown in the presence of GLY; GLY + NO—plants grown in the presence of GLY and weekly sprayed with SNP. Results are presented as mean ± standard deviation (SD) and result from the evaluation of at least three experimental replicates (*n* ≥ 3). Different letters indicate significant differences between groups (CTL, NO, GLY and GLY + NO) at *p* ≤ 0.05, according to the one-way ANOVA followed by Tukey’s post hoc test.

Parameter	CTL	NO	GLY	GLY + NO
Number of flowers per plant	13.3 ± 2.3 a	10 ± 3.4 ab	6.5 ± 1.0 b	10.0 ± 2.3 ab
Number of fruits per plant	8.0 ± 0.9 a	3.7 ± 1.2 b	3.6 ± 0.4 b	4.3 ± 0.8 b
Fruit fresh mass (g)	3.7 ± 1.1 a	3.0 ± 0.2 a	2.3 ± 0.2 a	3.2 ± 0.8 a

## Data Availability

The data presented in this study are available in this manuscript.

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
