# Peer review of "Foliar Application of Sodium Nitroprusside Boosts Solanum lycopersicum L. Tolerance to Glyphosate by Preventing Redox Disorders and Stimulating Herbicide Detoxification Pathways"

_plants, 2021, doi:10.3390/plants10091862_

Round 1

Reviewer 1 Report

The manuscript on foliar application of sodium nitroprusside boosts Solanum lycopersicum L. tolerance to glyphosate by preventing redox disorders and stimulating herbicide detoxification pathways is interesting and on a key topic that deserves attention and more research. The presentation, use of the English language and technical aspects are excellent, with clear use of proper methods and care in the presentation of the work. I am also very pleased to see that the analyses chosen are connected and complement each other, producing a solid comprehensive study reporting brand new findings and the mechanistic hypotheses explained. Of course that there are reports on this topic but what the authors are reporting is not just more of the same, but has new insights and original information.

I have only three comments: (1) Discussion - I think an extra comment/discussion will add information to the reader; (2) corrections in the reference list; (3) Figure style (suggestion).

  1. The authors have chosen some key antioxidant enzymes to measure and I agree with their choice, although many other could have been chosen. However, they opted to measure only total enzyme activity, which is fine, but the results would have been even better if they had also measured the activity of specific isoenzymes. For instance, statistically there have been changes in total enzyme activity in some conditions (see Fig 3b and Fig 4). These enzymes are well known for having multiple forms (isoenzymes), therefore, with total activity analysis some information is obtained, but it does not allow to discuss further insights into the role or whether such changes are due to a specific isoenzyme. But I understand that the data are important and have to be published in the contest reported, but I suggest to the authors to consider Native-PAGE in their future studies, whereas for this manuscript I think it is necessary to extra paragraph in section 3.4 indicating what I commented here, which is, the need for future analyses and importance of using native-PAGE to verify whether the changes observed in total enzyme activity can be attributed to any specific SOD, CAT, etc, isoenzyme. It is important to bear in mind that SOD, for instance, has several isoenzymes in tomato and they can be located in different cell organelles, therefore, changes in activity may be linked to specific metabolic roles of such organelles.

  1. The authors should double check the reference list for the name of the authors and their initials. For instance in refs 35, 49, 50, among others, I noticed initials missing for several authors. I also noticed some errors in the name of journals: see for instance refs 56 (J Biological Trace Element Research – delete “J”), 59 (J BMC Plant Biology – delete “J”), 61 (J Food Chemistry – delete “J”), 65 (J Horticulture Research – delete “J”) – it seems this identical error appears in several references cited among some other typographical errors (see ref 26). Finally, the name of some journals appear in capital letters (1st letter), whether others do not –be consistent in style and format. Please check and correct such imperfections since if they are published they will result in incorrect citations in the literature databases.

  1. Figures: The authors opted for Dark and Light bars to represent shoots and roots, respectively. It is fine, but the authors could consider instead using different bar colours (black, dark grey, light grey and white) for the treatments CTL, NO, GLY, GLY + NO, respectively, and indicate shoots and roots in the figure legends. I think that such modification will make easier to see and follow the results. But I leave that up to the authors.

Author Response

Dear Reviewer,

First of all, thank you very much for your comments and suggestions to improve the quality of our manuscript. Attached to this message, may you find a word document with a detailed response to all of your queries.

Best regards,

Cristiano Soares
Fernanda Fidalgo

Reviewer 2 Report

Comments and Suggestions for Authors

The manuscript titled “Foliar application of sodium nitroprusside boosts Solanum lycopersicum L. tolerance to glyphosate by preventing redox disorders and stimulating herbicide detoxification pathways” Manuscript ID: plants-1347957, by Soares et al., submitted for publication to Plants is an interesting study that presents a sustainable strategy to enhance the plant the tolerance against the most used herbicide worldwide, glyphosate (GLY). In the current study, the authors evaluated the potential protective role of nitric oxide (NO), provided sodium nitroprusside (SNP), in neutralizing the GLY- induced stress in tomatoes. Moreover, they deciphered the potential physiological and biochemical mechanisms behind the protective role of NO against GLY. The manuscript is well-planed, well-written, and presents sufficient data to support their hypothesis. However, I have very few comments and suggestions, just to improve the quality of the manuscript.

Comments and suggestions 

  • Introduction
  • Add a new paragraph (or at least a few sentences) about Sodium Nitroprusside (SNP) and the mechanism(s) of NO release, since this information is not known for everybody.
  • Results:
  • All tables (Table 1-3) and figures (Figures 1-6) should be self-explanatory. In other words, they should have self-explanatory captions that provide sufficient information to the readers without referring to the related text in the manuscript. The rules for composing the captions are the same as for composing the title of the paper. The reader should be able to look at a table or figure and by reading the caption know exactly what was done in that part of the experiment without having to read the text for an explanation. It should include information on important aspects such as description (the type of experiment), method or technique used (sample size, plant model used, conditions examined, etc.), and the result obtained (p-value). Moreover, they should also include an explanation of the features of a figure. This includes a description of symbols, patterns, lines, colors, non-standard abbreviations, scale bars, and error bars that are a part of the figure. You can exclude elements that are already described in the actual figure. Briefly, to fix this issue, you should:-
    1. Spell out the non-standard abbreviations within the caption of each figure/table. This includes the names of the treatments (CTL, NO, GLY, and GLY-NO), as well as the biometry names (AsA/DHA, GSH, TAC, TPC, O2.-, H2O2, MDA, SOD, and GST … etc).
    2. No need for supplementary material (Table S1-S3). It is better to add only the p-values within the figures/tables where it is appropriate.
    3. Be consistent in presenting units of measurement. For example, in figure 1 and all tables, you provided measuring units between parentheses (…), however, in the rest of the figures (Figures 2-5), you provided the units of measurement without parentheses. Please be consistent. Personally, I prefer providing the units of measurement between parentheses (…).

  • Throughout the manuscript, the authors tested four different treatments, (CTL, NO, GLY, and GLY-NO), however, when they studied the productivity-related characteristics (Table 3), they tested only three treatments (CTL, GLY, and GLY-NO) and the “NO” treatment was NOT included. Please be consistent and add the data for “NO” treatment.
  • In figure 6, I highly recommend separating the PCA-scatter plot from the PCA-loading plot for better presentation and to ensure clarity. Kindly, see below
  • Discussion

The discussion is very long (Approximately 6 pages) and diluted. The discussion section should inform readers about the larger implications of your study based on the results, but not every piece of finding. Highlighting these implications while not overstating the findings can be challenging, especially when you’re submitting to a journal that selects articles based on novelty or potential impact. Kindly, see the following link (https://plos.org/resource/how-to-write-conclusions/) for more discussion writing tips. The discussion section MUST be rewritten, shortened (no more 3 pages), and be more focused.   

4- Finally, although the manuscript is well-written and the language used was easy to follow, however, the manuscript should be carefully and deeply revised for grammar and English use, since some minor mistakes were found throughout the whole paper.

Author Response

(The authors gave the same response as above.)

Reviewer 3 Report

Please look at the file attached.

Most concerned about this article is the treatment effects that show no statistical relevance in the manuscript. Thus, more dosages of treatment could be needed to reveal the trend of treatment effect. Secondly, the article data expressed the parameters mainly on the base of tissue fresh-weight that could be dramatically influenced by tissue water content.

Author Response

(The authors gave the same response as above.)

Round 2

Reviewer 3 Report

Would you please provide the water content of shoot and root under treatments since proline contents are very sensitive to the water status of plant tissue?

Author Response

Please consider the attached file. 

Kind regards,

Cristiano Soares & Fernanda Fidalgo
